# What can Secondary Data Tell Us about Household Food Insecurity in a High-Income Country Context?

**DOI:** 10.3390/ijerph16010082

**Published:** 2018-12-29

**Authors:** Ourega-Zoé Ejebu, Stephen Whybrow, Lynda Mckenzie, Elizabeth Dowler, Ada L Garcia, Anne Ludbrook, Karen Louise Barton, Wendy Louise Wrieden, Flora Douglas

**Affiliations:** 1Health Economics Research Unit, University of Aberdeen, Aberdeen AB25 2ZD, UK; oejebu@abdn.ac.uk (O.E.); l.mckenzie@abdn.ac.uk (L.M.); a.ludbrook@abdn.ac.uk (A.L.); 2Rowett Institute, University of Aberdeen, Aberdeen AB25 2ZD, UK; stephen.whybrow@abdn.ac.uk; 3Department of Sociology, University of Warwick, Coventry CV4 7AL, UK; Elizabeth.Dowler@warwick.ac.uk; 4Human Nutrition, University of Glasgow, Glasgow G31 2ER, UK; Ada.Garcia@glasgow.ac.uk; 5Division of Food and Drink, Abertay University, Dundee DD1 1HG, UK; K.Barton@abertay.ac.uk; 6Human Nutrition Research Centre and Institute of Health & Society, Newcastle University, Newcastle upon Tyne NE2 4HH, UK; Wendy.Wrieden@newcastle.ac.uk

**Keywords:** food insecurity, food poverty, prevalence, household, food surveys, secondary data, Scotland

## Abstract

In the absence of routinely collected household food insecurity data, this study investigated what could be determined about the nature and prevalence of household food insecurity in Scotland from secondary data. Secondary analysis of the Living Costs and Food Survey (2007–2012) was conducted to calculate weekly food expenditure and its ratio to equivalised income for households below average income (HBAI) and above average income (non-HBAI). Diet Quality Index (DQI) scores were calculated for this survey and the Scottish Health Survey (SHeS, 2008 and 2012). Secondary data provided a partial picture of food insecurity prevalence in Scotland, and a limited picture of differences in diet quality. In 2012, HBAI spent significantly less in absolute terms per week on food and non-alcoholic drinks (£53.85) compared to non-HBAI (£86.73), but proportionately more of their income (29% and 15% respectively). Poorer households were less likely to achieve recommended fruit and vegetable intakes than were more affluent households. The mean DQI score (SHeS data) of HBAI fell between 2008 and 2012, and was significantly lower than the mean score for non-HBAI in 2012. Secondary data are insufficient to generate the robust and comprehensive picture needed to monitor the incidence and prevalence of food insecurity in Scotland.

## 1. Introduction

Household food insecurity (HFI) is a common problem for low-income households in high-income countries [1,2,3,4]. HFI exists when a household experiences the inability “to acquire or consume an adequate quality or sufficient quantity of food in socially acceptable ways, or the uncertainty that one will be able to do so” [5]. It has been empirically established to manifest itself involuntarily across four dimensions: (i) quantity and (ii) quality of food; (iii) psychological impacts; and (iv) socially unacceptable food and ways of obtaining food [6,7,8]. In high-income countries, food insecurity is more commonly characterised by chronic compromises in dietary quality and anxiety associated with accessing food. In contrast, low and middle income countries most generally experience acute and chronic episodes of food deprivation, hunger, and starvation [8]. Critically, for health and social care policy makers in high-income countries, the experience of food insecurity featuring poor diet quality leads to negative health outcomes i.e., cancer, stroke, cardiovascular disease, diabetes and obesity, and depression [8,9,10,11,12,13,14,15]. Much of the epidemiological evidence highlighting these associations has been generated in the North American context where routine capture of HFI data has taken place for some decades [16]. Nationally representative food security monitoring was established in the U.S. in 1995 with the highest recorded HFI prevalence observed during the global recession of 2008–2009 (14.5% of the population) [17]. While there has been an observed decline in those figures, they remain in the region of 12% of the population (15 million households) [18]. Canadian HFI prevalence runs roughly in line with the U.S. at 11–12% of the population found to have been food insecure since food security monitoring was established there in 1994 [19]. HFI prevalence is also higher in specific population subgroups including households with children, single-parent households and indigenous, black and Hispanic households [18,20], a pattern also observed in Australia and New Zealand [21,22]. It is also widely argued that HFI is an outcome of income insufficiency in relation to necessary household expenditures [20,22,23,24,25,26] and the decisions of policy makers [27]. In the late-2000s, HFI had re-emerged as a subject of public health, social policy, civic and political concern in Scotland and the rest of the UK. This is attributed to an increase in the numbers of people turning to emergency food supply centres (so-called food banks) for help with feeding themselves and their families [28,29,30,31,32]. Food bank use data became a de facto measure of food insecurity across the UK, largely through the high profile reporting of the Trussell Trust, which is one of the best known charitable organizations providing emergency food aid in the UK [33]. Yet the ability of emergency food supply centres to address HFI, or provide an insight into the nature and scale of food insecurity at the local and national level is problematic [6,17,18,34]. In similar international contexts, where it is possible to make comparisons with routinely collected food insecurity data, food bank use data tend to significantly underestimate the prevalence of food insecurity [35,36]. This presents challenges for policy makers tasked with the development, implementation, and evaluation of policy measures aimed at addressing the problem [37]. It is particularly problematic for governments and policy makers given that food banks, as a de facto public policy response to the problem, are unable to meet local demand for food assistance for a variety of inherent resource constraints, unless food is rationed by restricting access for those who want and are able to access them [34,38,39].

It was in this context, and in the absence of any purposively collected and experiential household food insecurity population survey data, that research was undertaken in Scotland, in 2014, to explore the nature and prevalence of HFI. This paper reports on the study component that screened and analysed relevant existing secondary data, with the specific aim of finding out what could be determined about the nature and prevalence of HFI in Scotland, as defined above [5]. The study was also guided by a particular focus on the years preceding the UK economic recession (2008 to 2009) [40] and the period of rapid increase in food prices (2007 to 2012) [41]. Additionally, the analysis included a comparison of the diets of households at risk of food insecurity compared to those at less risk. The paper discusses what analysis of existing data sources is able to reveal (or not) about two of the four defined dimensions of the experience of food insecurity in the Scottish context, i.e., dietary quantity and quality, and reflects on the absence of data on psychological and social experiences, and the implications of this research for future policy making in this area.

## 2. Materials and Methods

The secondary data analysis proceeded in three stages.

### 2.1. First stage—Scoping and Data Source Selection

The first stage involved a scoping, consultation and decision making process to identify suitable Scottish data sources to explore food insecurity patterns covering the 2008 UK recession. It was at this stage also that consideration and agreement was reached about the ‘at risk’ household income threshold that would be used during the analysis, to take account of the lack of a UK food insecurity measure [37].

The threshold level agreed for identifying those at risk of food poverty was equivalised net household income of less than 60% of the median value [42,43], which is the commonly used measure of poverty in the UK. Household income is equivalised to take account of household size and composition (including numbers of adults and children, and their ages [44]). Households below the threshold are referred to as households below average income (HBAI). Households with income above this threshold are referred as non-HBAI. Income was measured before housing costs were deducted [43].

Using the aforementioned HFI definition [5], datasets were identified where relevant variables for analysing HFI trends and prevalence in Scotland were available. Six potentially suitable datasets were identified, with four being rejected. These were, (i) the General Lifestyle Survey [45] (excluded due to difficulties gaining timely data access); (ii) the European Union Statistics on Income and Living Conditions [46] (Scotland is not identified as a separate UK region); the Family Resources Survey [47] (insufficient information on food purchase data); and Kantar Worldpanel [48] (which includes very few low income households and insufficient information to calculate equivalised income). The two remaining datasets, the Living Costs and Food Survey (LCFS) [49] and the Scottish Health Survey (SHeS) [50], were used for this analysis.

The LCFS is an annual stratified random sample survey conducted by the Office for National Statistics. It includes approximately 500 households in Scotland. The survey collects information on household spending patterns from diaries of daily expenditure recorded over a 2-week period. LCFS data from 2007 and 2012 were used for the present study. Variables include weekly household expenditure on food and non-alcoholic drinks brought home, eaten away from home (e.g., at a restaurant or hotel) and take-away items. Weekly food expenditures were adjusted to 2013 prices using the food and non-alcoholic drinks Consumer Price Index (CPI). Food expenditure-to-income ratios were calculated by dividing weekly household food expenditure by weekly equivalised household income using the McClement equivalence scale [44]. Household income was adjusted for inflation using the overall 2013 CPI.

From 2008, the SHeS was conducted annually. It contains information on the prevalence of different health conditions and health-related behaviours, including dietary intake collected on alternate years. Data are collected at an individual (both adults and children) and a household level. It includes around 6500 individual observations in 2008 and 4800 in 2012. Information on usual daily food eating patterns (type of food and frequency of consumption) are provided. SHeS also collects data on annual household income and converts this value to equivalised income using the McClement equivalence scale.

Both datasets were weighted using available sampling weights, which adjust for non-response and to match the population distribution [49,51].

### 2.2. Second Stage—Prevalence Estimation and Sub Group Analysis

Using both LCFS and SHeS, the second stage involved estimating the numbers of households at risk of being in food poverty, and investigating how prevalence had changed over time, and comparing these changes with data from the Family Resources Survey. Information on household income and prevalence of HBAI from the Family Resources Survey [47] was included to place the Scottish data in the context of data for the whole of the UK. The FRS (Family Resources Survey) is the most comprehensive survey of household financial circumstances using a large sample of UK households, and is the government source for poverty level analyses.

Using LCFS, this stage also included analysis of food expenditure (£) and food-to-income shares (%) between lower and higher income households. Mean weekly expenditure on food and non-alcoholic drinks, and their corresponding values in terms of percentage by food group based on the Eatwell Plate [52] were also calculated. Two-tailed independent t-tests were used to compare mean food expenditure (overall and by food group) and food-to-income shares between HBAI and non-HBAI, and to compare percentage DQI score (see below) between HBAI and non-HBAI and over time.

### 2.3. Third Stage—Dietary Quality Assessment and Analysis

The third stage involved assessing differences in overall diet quality between those considered at risk, and those not at risk, of being in food poverty. Dietary recommendations are based on the amounts of foods consumed, whereas food and drink are recorded in the LCFS “as purchased”. These were adjusted to “as consumed” values per person by accounting for food waste, and food preparation and cooking weight changes [53,54,55]. Nutrient intakes were calculated using the LCFS food composition database [56].

Within the SHeS, an Eating Habits Module (EHM) assesses consumption of a simple list of foods that are relevant to the Scottish Dietary Goals [57]. The EHM focuses on frequency of consumption of specific foods and was not designed to quantify amounts of foods or nutrients consumed, or meal patterns. It is not possible to assess nutrient intake, household food practices, meal patterns or experiences of the stability of the household food supply from the EHM. The EHM consists of two sections, the first being a series of questions on the consumption of food and drink items to gather information on general eating habits using a food frequency questionnaire methodology. The second assesses fruit and vegetable intake by a 24 h recall method using “everyday” food portion terms (such as tablespoons, cereal bowls and slices). Information on the number and type of fruit and vegetables eaten by respondents the day prior to the interview was used to compare the percentage of individuals in HBAI and non-HBAI reaching the 5-a-day goal for portions of fruit and vegetables.

A more comprehensive measure of diet quality, using the Diet Quality Index (DQI) devised by Barton and colleagues [58,59], was also used to calculate scores for SHeS and LCFS. Diet quality indexes are frequently used to summarise how well an individual’s diet compares to a collection of dietary recommendations, based on foods and nutrients considered to be important to health [60]. For example, adherence to the Dietary Guidelines for Americans can be assessed using the Healthy Eating Index, which has been shown to be a valid and reliable index of diet quality [61].

Diet Quality Index scores were calculated for the LCFS (2007 and 2012) and SHeS (2008 and 2012). For the LCFS data, DQI scores were calculated for a combination of foods (fruit and vegetables, fish, and red meat) and nutrients (percentage energy from fat and saturated fat, sugar and complex carbohydrates, and fibre) [58]. For the SHeS data, DQI scores were calculated from seven food components: oil-rich fish; red meat and processed meat; starchy foods; fibre in foods; sugary foods; fatty foods; and fruit and vegetables. The difference in food items used to calculate the DQI is because of the variations in dietary information available from the LCFS and SHeS. Absolute values for the DQI from the two surveys are therefore not directly comparable and have been expressed in the results as a percentage of the maximum possible score for each survey. Higher scores indicate greater adherence to dietary guidelines.

The proportions of food groups contributing to each diet were estimated using the Eatwell Plate recommendations. In the UK, the Eatwell Plate [52] (now updated and renamed the Eatwell Guide) was developed for representing nutrient intake information in a picture format to make dietary recommendations easier for consumers to understand. The Eatwell Plate is a pie-chart diagram consisting of five food group segments, the recommended proportions of which are based on the dietary reference values for the population. The five groups being: 1. bread, rice, potatoes, pasta and other starchy foods (starchy, which should make up around 33% of the diet), 2. fruit and vegetables (F&V, 33%), 3. milk and dairy foods (dairy, 15%), 4. meat, fish, eggs, beans and other non-dairy sources of protein (protein, 12%) and 5. foods and drinks that are high in fat or sugar, or both (HFHS, 8%).

Statistical analyses were carried out using STATA Version 13 (StataCorp LP, College Station, Texas, TX, USA) and SPSS Version 22 (SPSS/IBM Corp, Armonk, New York, NY, USA).

## 3. Results

### 3.1. Households at Risk of Food Insecurity—Prevalence Estimates

Table 1 shows the number and proportion of HBAI estimated in the LCFS, SHeS, and from the FRS.

Prevalence estimates from the LCFS and SHeS are similar. In contrast, the results using LCFS data are weighted to adjust for non-response and to match population distributions, and give higher levels than those in the FRS. However, all estimates show an apparent decline in the prevalence of HBAI between 2007 and 2012.

Table 1 also shows the mean weekly expenditure on food (including food eaten away from home and take away food) and non-alcoholic drinks using LCFS. Results are displayed for HBAI and non-HBAI for 2007 and 2012, respectively. HBAI spent less actual money per week on food than non-HBAI (Table 1) (*p* < 0.001 in both years), but the proportion (%) of equivalised household income spent on food was approximately twice the proportion spent by non-HBAI (*p* < 0.001 in both years). There is a slight decrease in both food expenditure and the share of food expenditure to income from 2007 to 2012 for HBAI. However, there is a bigger drop in food expenditure by non-HBAI combined with an increasing share of income being spent on food. This suggests that non-HBAI had more discretion to reduce food spending in the face of declining real incomes during the period of recession.

### 3.2. Dietary Analysis and Assessment

Table 2 reports the (mean) weekly expenditure of HBAI and non-HBAI, by Eatwell Plate food group, as well as the corresponding percentage share of income (%). In contrast to the results in Table 1, any other food expenditure (e.g., food eaten away from home and take-away food) are excluded from this calculation.

For each food group, while HBAI spend significantly less of their weekly income in pounds (£), they spend proportionately (%) more in comparison to non-HBAI (*p* < 0.001 for each food group in both years). There is no statistically difference of expenditure between HBAI and non-HBAI for Non-alcoholic drinks in year 2012 and ‘Other’ food (for both years).

Non-HBAI households spend more on both ‘healthy’ food (fruit and vegetables) and ‘unhealthy food’ (foods high in fat and sugar (HFHS)), suggesting that poor dietary choices are not necessarily determined solely by spending power. Meat and other sources of proteins, and HFHS represent the largest share of food expenditure in both HBAI and non-HBAI alike. Noticeably, fruit and vegetables constitute the third largest food expenditure in both household types.

Figure 1 shows the percentage of SHeS individuals from HBAI and non-HBAI by number of portions of fruits and vegetables consumed on the day prior to the interview. In both years, there is a marked difference in the proportion of individuals reaching the 5-a-day target between those in HBAI and non-HBAI (14% and 32%, *p* < 0.001 in 2008 and 12% and 21%, *p* < 0.001 in 2012). Respondents in HBAI were more likely to report consuming no, or only one portion of, fruit or vegetables the previous day compared to their non-HBAI counterparts.

The proportion of individuals in HBAI who reported eating no fruit and vegetables the day prior to the interview was higher in 2012 (18%) than in 2008 (11%). Nevertheless, 14% of individuals from HBAI reported eating five or more portions of fruits and vegetables in 2008; this proportion fell to 11% by 2012. However, there was little change in the proportion of individuals from non-HBAI eating five or more portions of fruits and vegetables over time (22% and 21% respectively).

Examination of the DQI scores calculated from the LCFS revealed no significant differences between HBAI and non-HBAI households for percentage DQI score (35.1% and 36.5%, *p* = 0.327 in 2007, and 36.2% and 34.7%, *p* = 0.506 for 2012) (Table 1). Examination of the DQI score based on the SHeS data showed the overall mean percentage DQI scores were similar in 2008 for HBAI and non HBAI (50.4% and 51.6% respectively, *p* = 0.196). However, by 2012, the overall percentage DQI score was significantly lower for HBAI than non-HBAI (48.5% and 51.6% respectively, *p* = 0.001).

## 4. Discussion

This study aimed to establish what could be determined about the nature and prevalence of household food insecurity in Scotland from secondary data. The scoping study established that it was possible to gain only a partial picture of HFI in the Scottish context, allowing a focus on quantity and quality of diets. Therefore, the subsequent analyses centered on an exploration of food expenditure-to-income shares and levels of food expenditure, and a dietary quality analysis of foods purchased and reported as consumed. The analysis focused on HBAI and non-HBAI households, over the period following the recent economic recession. Consequently, this discussion proceeds in two parts, focusing firstly on the findings of the ‘partial picture’ data able to be accessed and analysed, and secondly, reflecting on the policy and practice implications arising from the lack of routine capture of psychosocial domain data of the experience of household food insecurity in the population.

### 4.1. Household Expenditure and Food and Nutrition Security

This analysis revealed that low income households in Scotland have continued to allocate a greater income share (%) to food over the period following the recession, with food expenditure a particularly prominent component of all household expenditure, compared to wealthier households. This is consistent with the expenditure patterns reported for the whole of the UK [62]. Findings from the current study suggest that the HBAI group had less margin to reduce food expenditure than their wealthier counterparts. For, at the same time that there was a reduction in food spending for this group, the income share devoted to it increased, as it did for non-HBAI. These findings also align with the current position, as more recent UK figures suggest that increases in food prices have continued to exacerbate the situation of low-income households due to the larger income share they devote to food expenditure, compared with higher income households [63]. Given the relative stasis of, and in many cases decline in, household incomes in low income households in the UK [64], the net effect on those lower income households is that they probably have less available income to spend on other essential household items.

Indeed, although there appear to be few differences in the diets of HBAI and non-HBAI when the frequencies of consumption of all key food groups were compared using the DQI, in both the LCFS and SHeS data, the consistent exception is in fruit and vegetable consumption. Fruit and vegetable consumption is commonly used as proxy indicator of diet quality. Lower income households have lower expenditure on fruit and vegetables than do higher income households (LCFS); and their self-reported consumption was lower compared with non-HBAI (SHeS). These results from the SHeS partially support previous research indicating that the quality of dietary intake is poor across all income groups in Scotland, but tends to be worse in the poorer households [65], and that these effects have worsened over time. Overall, the indicators of HFI and the subsequent analysis used in this study are innovative and could be adapted in other studies to bridge the gap in the literature.

As this research was designed to examine the usefulness of existing relevant data sources in enabling the characterization of HFI in Scotland, it is important to note that the SHeS is designed to estimate frequency of consumption rather than provide estimates of the amounts of foods consumed. This makes interpretation of dietary quality difficult, since the Scottish Dietary Targets are based on amounts of food groups consumed. It is conceivable that those on the lowest incomes have already made all possible adjustments to expenditure and cannot further reduce spending, as argued above [62]. This finding might also be explained by the relatively short shelf life of fresh fruit and vegetables, which often makes such items relatively unattractive to purchase compared to other less perishable items for those whose room for financial manoeuver is more limited.

Consequently, when considering and interpreting these findings, it is important to note that virtually no attention had been paid to people’s lived experience of food insecurity in Scotland until very recently. No study has focused on the experience of food (in)security in the context of other necessary household expenditures [66], that might explain these patterns of difference in diet quality and, for example, expenditure on fruit and vegetables. Why these patterns persist, despite longstanding educational campaigns exhorting the benefits of healthy eating, has also not been investigated. Where empirical research (in high income country contexts) has been conducted to investigate the direct perspectives and motivations of different socioeconomic households regarding “healthy” food provisioning and meal preparation (as opposed to drawing inferences from dietary pattern data), low income/food insecure households are no less likely than their wealthy counterparts to express a desire to consume healthy food [67]. Indeed, there is some evidence to suggest that very low-income households possess significant food knowledge and skills, and employ multiple strategies with the aim of feeding the family nutritious foods [68,69,70,71,72]. The evidence presented above regarding income shares devoted to food in Scotland in recent years suggests that the capacity for very low income households to be able to take action according to their aspirations and preferences could be constrained; which is consistent with the views of health, social care and third sector practitioners supporting economically and social vulnerable groups in Scotland [73].

The analysis also revealed an apparent decrease in the proportion of households whose household income fell below 60% of the median income value between 2007 and 2012, which some may construe to mean a subsequent decline in the numbers of households affected by food insecurity. The poverty threshold used in this study was based on the median income of each sample survey, and not the UK or Scottish median income; this might partially explain disparities in the percentage of HBAI within the FRS. In addition, since the “HBAI at risk marker” is based on a single threshold each year, any change in the overall nature of the distribution of equivalised household income between survey years could affect the position of the median relative to the mean equivalised income. This could influence the proportion of households above and below the poverty threshold level. Median income was likely to be falling in real terms due to the recession. Existing Scottish Government surveys and reports provide a more thorough and complete view of the prevalence of poverty per se [74,75].

In addition, the food poverty threshold used in this study (<60% of the median equivalised household income) masks the experiences of households with considerably lower incomes, which are also likely to be underrepresented in the datasets used in this analysis. Individuals whose household income is below 50% or 40% of the UK median income are considered as living in severe or extreme poverty, respectively [74]. Moreover, those just above or below this threshold may be quite similar, and experience similar financial difficulties from increased household costs.

Furthermore, households living on remote Scottish islands are not included in the LCFS, for data collection cost reasons. This is an important omission, given that incomes are known to be lower in rural areas, and that those living in remote and rural areas in Scotland need to spend “10–40% more on everyday requirements than elsewhere in the UK” [76]. In the LCFS, sampling variability was also affected by a higher non-response rate of households whose head had no post-school qualification or was in a manual social class group [56]. In the SHeS, deprived areas were over-sampled however [56]. In both LCFS and SHeS, weights were used to reduce the effect of non-response bias so that the sample distribution matches the population distribution in terms of region, age group and sex [49,77].

However, international evidence derived from jurisdictions where food insecurity is routinely monitored shows that while household income is an important determinant, it does not fully explain the circumstances of all those who are food insecure. Housing costs (i.e., mortgage, rent, fuel and insurances), and other necessary household expenditures (such as travel and debt) have been shown to significantly contribute to the observed prevalence of food insecurity [78,79].

### 4.2. Study Implications

While the current analysis revealed some important and worrying patterns of population dietary deterioration in HBAI households at the same time as they are apportioning more and more of their income to food expenditure in Scotland, it also revealed some crucial gaps in the available data. For example, it was not possible to show which groups of people were most at risk from food insecurity. As highlighted previously, in other high income jurisdictions it has been possible to identify specific household types that are more at risk of severe and enduring HFI, and with it, the theoretical possibility of targeted policy interventions [18,20]. The need for this data has been brought into sharp relief in recent times in the UK as a recently published report by the UN Rapporteur for Extreme Poverty and Human Rights expressed great concern for the working poor, female-headed households, children and those living with disabilities, pensioners, asylum seekers and refugees and those living in rural poverty as most at risk of extreme poverty in the UK at the present time [80].

An analysis of children living in households at risk of HFI was not possible either. Such analysis would be very informative considering the latest finding of a UNICEF report [81]. It revealed that in 2014, 19.7% children aged 0–17 were living in HBAI in the UK. Whilst this stands below the average of developed countries (21.0%), it is an indication of the extent to which HFI affects vulnerable groups.

Another important gap revealed was the lack of data that could provide a picture of the people and groups who were affected by uncertainty/anxiety associated with being able to afford to feed oneself or the family. Nor was it possible to determine the duration and frequency of these types of experiences at the household level. Based on U.S. observational studies [6], markers of food deprivation are regarded as more sensitive than income-based measures alone, in capturing not only the material aspects of deprivation (largely caused by income poverty), but also its combined biological and psychosocial effects on health and well-being. A number of international studies have also concluded that the experience of HFI is likely to be impacting human health as much through ‘non-nutritional’ mechanisms, such as worry, anxiety, feelings of deprivation, and social isolation, as through nutritional routes [6,82,83,84].

Indirect measures of HFI, such as food availability, purchasing power, consumption patterns and anthropometric measures were deemed to be insufficient for HFI monitoring and the evaluation of interventions intended to address it as far back as the late 1970’s [85]. Within other high-income country contexts, where routine HFI monitoring takes place, a now significant body of research has linked HFI with negative physical and psychological health consequences [16,68,86,87]. It is also known to impair chronic condition management [88,89] and is independently associated with increased health care use and costs [90].

Therefore, it has been encouraging to witness the discussion and policy shifts in recent years highlighting the need for routine monitoring of HFI across the UK. However, there is still no agreement about the means and measures by which this should be done, with policy differences emerging within the different nations of the UK regarding these [91,92]. For example, the Scottish Government have recently accepted the main recommendations of the Independent Working Group on Food Poverty in Scotland [93] and have introduced a HFI measure (a derivative of the UN Food Insecurity Experience Scale) [94] into the SHeS. The SHeS operates on a continuous annual reporting basis, and provides sufficient data for each individual health board area in Scotland to understand their population health dynamics over time, and has the potential for data linkage with other population data sets including disease registers [95]. The SHeS is specifically funded to monitor population health outcomes and trends. Therefore, the inclusion of a HFI measure in this survey should make it possible to determine the effectiveness of policy interventions intended to address HFI, as well as understanding the role of HFI in the poor health outcomes of the Scottish population. The benefits of placing this measure here, and with it the potential routine capture the more multi-dimensional HFI experience, are manifold. Firstly, it provides the facility to assess and monitor food insecurity experience for different subgroups (e.g., geographic location, age, ethnicity, household type, occupational status and health status). Secondly, embedded HFI monitoring in such a survey enables data linkage with other population data sets including disease registers, and therefore enables better scrutiny of the impact of food insecurity experience on population health outcomes [95]. Thirdly, the inclusion of a such HFI measure also provides the facility to monitor prevalence and severity (if not chronicity), and with it the potential to develop better understanding of the role different HFI experiences (in terms of nature and severity) has in relation to health outcomes within the Scottish population, something that was beyond the scope of this study. Fourthly, it should also provide a robust means to determine the effectiveness of policy interventions intended to address HFI. Indeed, it is important to stress the benefits of introducing and retaining such a measure compared to the HFI indicator used in Europe, which uses a more unidimensional indicator that is based primarily on the prevalence of the household’s inability to afford meat/fish/poultry (or a vegetarian equivalent) every second day [25]. Fifthly as the SHeS survey routinely also captures household income data, it should be possible to monitor and model changes in HFI prevalence in the context of changing national and household economic circumstances and social policy changes, something one-off cross-sectional studies cannot undertake. There have been calls for a similar type of measure to be introduced in England and Wales [96].

In 2003, the UK Food Standards Agency used a piloted questionnaire based on the USDA (United States Department of Agriculture) experience to investigate household food security in a UK-wide survey of diet and nutrition in low-income households [97]. A similar questionnaire was used in the Food Standard Agency’s ‘Food and You’ consumer survey in England in 2016 [98]. However, this seems unlikely to offer the same depth and functionality as the data collected by the SHeS going forward. In addition, none of the measures mentioned above include questions about children’s food security status. This is a serious omission given the reported levels of child poverty and so-called ‘holiday hunger’ in the UK, relating to the important role that is played by provision of free school meals to low income families during term time [99], and the associated poor child health and educational outcomes that have been observed in international contexts where children’s HFI status is captured and recorded [100].

It is important also to acknowledge that, while routine food insecurity measurement offers the potential to characterise and monitor HFI prevalence more comprehensively, and provides a means to evaluate policy and programmatic interventions aimed at addressing it, challenges will remain in capturing the experiences of homeless and destitute individuals and families, and representing them in secondary data. Routine HFI monitoring and evaluation endeavors must therefore also include direct qualitative engagement with those highly vulnerable groups and the services and/or agencies responsible for their care and support, to inform and evaluate social, economic and health policy changes intended to address HFI [73]. In addition, whilst periodic episodes of absolute food deprivation are a public health and ethical concern of any country, it is also the case that the less severe experience of food insecurity resulting in chronic dietary quality compromises over time, is likely to be the more prevalent experience in high income countries like Scotland. This is an important distinction to be able to capture as accurately as possible as chronic food insecurity experience is considered to be as damaging to health and well-being over the long term as periodic deprivation experiences [68] and this phenomenon needs more research attention in HFI monitoring work in the UK and elsewhere than it currently receives. Related to this is the need in the UK to develop a better understanding to the impact of food insecurity on chronic or long-term health condition management in the UK [73]. Having a clearer picture of the severity and chronicity of people’s HFI experience as well as national and subnational prevalence would assist researchers and policy makers to develop more insight into this overlooked health care issue, and create better informed policy responses to address HFI in the Scotland and elsewhere in the UK.

## 5. Conclusions

This study provided a partial picture of the prevalence of HFI in Scotland. It revealed that low income households have been consuming a diet that has further deteriorated in nutritional quality over the period following the recession and have been spending a significantly higher proportion of their household incomes compared to wealthier households. Other important dimensions of HFI are unavailable for scrutiny for monitoring and evaluation purposes. Additional or alternative measures to identify at-risk households are required to inform the development and evaluation of policy and programmatic interventions intended to address the problem. Routine and systematic monitoring would not only enable HFI incidence and prevalence to be better characterised, but would enable the relationship between household-level problems of food insecurity and changing social and economic conditions to be monitored and understood. In order to develop, implement and evaluate social and public health policy interventions intended to reduce the numbers of households affected by food insecurity, the routine capture of household food insecurity data suitable for Scottish and UK population health monitoring is urgently needed.

## Figures and Tables

**Figure 1 ijerph-16-00082-f001:**
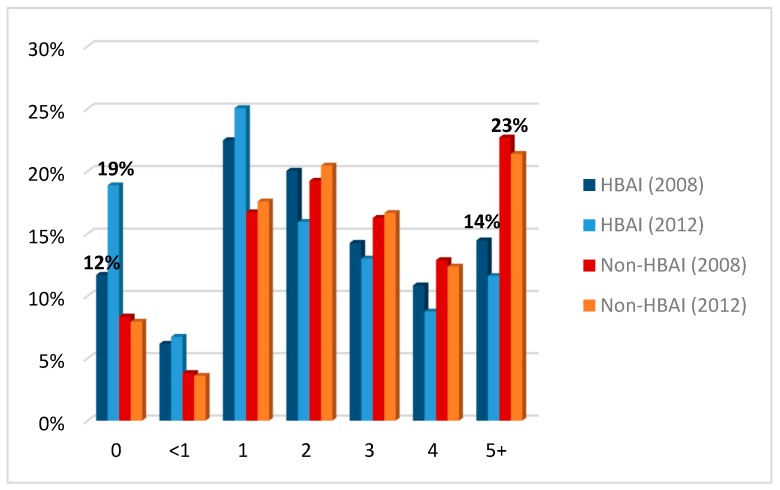
Percentage of individuals from HBAI (households below average income.) and non-HBAI by fruit and vegetables consumption (number of portions on the day prior to the interview). Source: computed by the authors based on SHeS (2008 and 2012)—weighted data.

**Table 1 ijerph-16-00082-t001:** Prevalence of households below 60% median income in Living Costs and Food Survey (LCFS), Scottish Health Survey (SHeS) and the Family Resources Survey (FRS), along with mean weekly expenditure and measures of diet quality.

Year of Survey	LCFS		SHeS		FRS	
	2007	2012	2008	2012	2007/8	2012/13
Scottish observations (households)	501	483	3567	2697	NA	NA
Monthly equivalised median household income (£)	£2079	£2039	£1842	£1954	£1699	£1907
Poverty threshold * (£)	£1247	£1223	£1105	£1172	£1019	£1144
Percentage of HBAI (Number)	23.4% (117)	18.9% (92)	26.3% (940)	23.4% (632)	17%	16%
Percentage of non-HBAI (Number)	76.7% (384)	81.1% (391)	73.6% (2627)	76.6% (2065)	NA	NA
Weekly expenditure on food and drinks (£)—HBAI ^$$^	£54.08 [48.30–59.86]	£53.85 [45.11–62.6]	NA	NA	NA	NA
Weekly expenditure on food and drinks (£)—non-HBAI ^$$^	£102.14 [95.41–108.87]	£86.73 [81.43–92.02]	NA	NA	NA	NA
p-values of mean food expenditure between HBAI and non-HBAI	*p* < 0.001	*p* < 0.001				
Weekly expenditure on food and drinks (% income)—HBAI ^$$^	30.7% [22.36–39.04]	29.4% [23.19–35.53]	NA	NA	NA	NA
Weekly expenditure on food and drinks (% income)—Non-HBAI ^$$^	14.1% [13.14–15.06]	15.5% [14.38–16.59]	NA	NA	NA	NA
*p*-values of food-to-income ratios between HBAI and non-HBAI	*p* < 0.001	*p* < 0.001				
DQI score (%)—HBAI	35.1% [31.9–38.3]	36.2% [31.8–40.5]	50.4%	48.5%	NA	NA
DQI score (%)—non-HBAI	36.5% [34.7–38.4]	34.7% [33.0–36.5]	51.6%	51.6%	NA	NA
*p*-values of DQI scores between HBAI and non-HBAI	*p* = 0.327	*p* = 0.506				

* 60% of monthly equivalised median household income (£). ^$$^ weekly food includes grocery shopping, non-alcoholic drinks, food eaten away from home (e.g., at a restaurant or hotel) and take-away food. Confidence interval into [brackets]. HBAI: households below average income.

**Table 2 ijerph-16-00082-t002:** Mean weekly expenditure * (£ and % of income) of HBAI (households below average income.) and non-HBAI, by food group.

Food type	2007				2012			
	HBAI		non-HBAI		HBAI		non-HBAI	
Starchy food	£5.91 [5.16–6.67]	2.49% [1.80–3.17]	£8.06 [7.45–8.66]	0.87% [0.79–0.94]	£5.28 [4.24–6.31]	3.01% [2.06–3.95]	£6.93 [6.42–7.43]	1.25% [1.13–1.37]
	*p* < 0.001 *p* < 0.001				*p* = 0.005 *p* < 0.001			
Fruits and vegetables	£6.85 [5.73–7.97]	2.54% [2.00–3.09]	£12.83 [11.82–13.85]	1.31% [1.20–1.41]	£7.22 [5.59–8.86]	3.70% [2.81–4.59]	£10.83 [9.98–11.69]	1.86% [1.69–2.03]
	*p* < 0.001 *p* < 0.001				*p* < 0.001 *p* < 0.001			
Milk and dairy	£5.16 [4.47–5.85]	2.09% [1.64–2.53]	£7.75 [7.02–8.46]	0.83% [0.75–0.90]	£5.10 [4.26–5.94]	2.71% [2.13–3.28]	£6.28 [5.84–6.73]	1.09% [1.01–1.1]
	*p* < 0.001 *p* < 0.001				*p* = 0.015 *p* < 0.001			
Meat and protein	£12.22 [10.53–13.92]	4.20% [3.53–4.87]	£19.87 [18.35–21.38]	2.09% [1.92–2.25]	£12.08 [9.67–14.48]	6.52% [4.82–8.23]	£18.2 [16.70–19.71]	3.15% [2.86–3.44]
	*p* < 0.001 *p* < 0.001				*p* < 0.001 *p* < 0.001			
HFHS	£10.72 [9.26–12.19]	4.71% [3.07–6.36]	£19.39 [17.69–21.10]	2.07% [1.88–2.27]	£11.78 [9.80–13.76]	5.95% [4.91–6.99]	£17.12 [15.81–18.44]	3.05% [2.75–3.35]
	*p* < 0.001 *p* = 0.0018				*p* < 0.001 *p* < 0.001			
Non-alcoholic drinks	£1.07 [0.84–1.30]	0.37% [0.28–0.45]	£1.72 [1.46–1.98]	0.18% [0.15–0.21]	£1.11 [0.78–1.44]	0.71% [0.25–1.17]	£1.46 [1.22–1.71]	0.25% [0.21–0.30]
	*p* < 0.001 *p* < 0.001				*p* =0.095 *p* = 0.053			
Other food	£0.08 [0.01–0.14]	0.03% [0.004–0.05]	£0.13 [0.003–0.23]	0.02% [0.00–0.03]	£0.05 [0.01–0.09]	0.02% [0.006–0.04]	£0.14 [0.08–0.20]	0.03% [0.02–0.04]
	*p* = 0.3756 *p* = 0.4632				*p* = 0.014 *p* = 0.856			
Total *	£42.01 [37.40–46.65]	16.42% [13.26–19.60]	£69.73 [65.06–74.41]	7.37% [6.82–7.90]	£42.62 [35.77–49.47]	22.62% [18.05–27.18]	£60.96 [57.24–64.70]	10.68% [9.99–11.37]
	*p* < 0.001 *p* < 0.001				*p* < 0.001 *p* < 0.001			
Observations	117		384		92		391	

* Excludes spending on food eaten away from home and takeaway food. Confidence interval into [brackets]. *p*-values represent the statistical differences between HBAI and non-HBAI for (i) mean weekly expenditure and (ii) food-to-income ratios respectively. Source: computed by the authors based on LCFS 2007 and 2012. Weighted and adjusted for inflation.

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
