# Peer review of "What can Secondary Data Tell Us about Household Food Insecurity in a High-Income Country Context?"

_ijerph, 2018, doi:10.3390/ijerph16010082_

Reviewer 1 Report

Thank you for the opportunity to review. I think this is a worthy piece of work that makes important and well-justified observations throughout in respect of the implications for nutritional quality of diets, proportionality of household spend on food expenditure and how this maintains and/or worsens over time. The limitations are clearly articulated and there is very good articulation of research ideas into the future in respect of linked datasets.

I question whether there is some reframing necessary so that the paper does not claim to state household food insecurity prevalence in the absence of an agreed indicator. I would suggest that minor additional proofreading is required to identify and correct the occasional missing word and/or punctuation. 

Author Response

Dear Reviewer 

Thank you for your time and consideration of our manuscript. I have appended our responses to all three of our reviewers so that you can see the genesis for all the changes we have made our manuscript.

Best wishes

Flora Douglas (on behalf of all authors)

Reviewer   1

Author’s   response

Thank  you for the opportunity to review. I think this is a worthy piece of work   that makes important and well-justified observations throughout in respect of   the implications for nutritional quality of diets, proportionality of   household spend on food expenditure and how this maintains and/or   worsens over time. The limitations are clearly articulated and there is very   good articulation of research ideas into the future in respect of linked   datasets.

1.       We are grateful for your time   spent reviewing our paper and appreciate your assessment and observations.

I   question whether there is some reframing necessary so that the paper   does not claim to state household food insecurity prevalence in the absence   of an agreed indicator. I would suggest that minor additional   proofreading is required to identify and correct the occasional missing word   and/or punctuation. 

2.       We have altered the title to try   to address this concern and hope that this modification and the further   additions and changes made to our paper in the round, that are detailed   below, will help address this concern.

We have thoroughly checked the paper again   and have spotted and sorted a number of minor typographical errors that the   reviewer quite correctly has drawn our attention to.

Reviewer 2 Report

Thank you very much for this innovative and interesting paper.  Some changes need to be made but the approach is novel and well done.

 Can secondary data tell us about household food 2 insecurity in a high-income country context?

Thank you for allowing me to review this contribution to the Scottish food insecurity knowledge base. I think that this paper is important and provides a great deal of information, which may not be directly evidence of food insecurity, but is certainly evidence of hardship and patterns of consumption associated with that.

I am very puzzled by the conclusion as it doesn’t really fit with the analysis or the title.  First of all, the use of the data in the study confirms many other study findings and is a valuable source of information about low income households in Scotland – so in that sense yes it can tell us something about food insecurity if we accept the premise that there is a relationship between the two (lots of literature on this).

While the definition of what makes a food insecure household may not be broad enough as it is not only low income that determines food insecurity, there is plenty of good information in this paper that suggests they are tapping into food insecurity (which the authors briefly reference in their introduction and discussion).

The introduction needs to be increased to summarise what is know from the international literature or food insecurity in high income countries. This paper is too narrowly focused on Scotland which is fine for the actual study itself but the introduction should be a broad base of information about the subject matter and it is not.  Increasing this will make the discussion easier as it too lacks enough connection with what is already known and focuses too much on just Scotland – the aim of research is to be able to be generalised where possible and much of the international literature is.

You also have used innovative methods and this is a real contribution to the knowledge base and may be able to be used (or adapted for use) by other studies.  This is a positive part of your study and needs to be acknowledged.

The discussion appears to be more about what is lacking and use as a platform for reform than what was added to the knowledge base.  The way in which this is written begs the question why do it at all if it is so inadequate??  Obviously this in not what the authors meant (I hope).  I would suggest adding at least a paragraph showing just how much extra information they have successfully put together about 1) low income households in Scotland and 2) their consumption habits.

The biggest omission and one that absolutely must be addressed is around the lack of any explanation of the statistics used in the production of the results.  The methods used were innovative and done well but where is the information about the statistical analysis?  The following additional information is required:

1)      While the description of how the measures were developed is excellent it is not statistical analysis.  Please say what statistical analysis you conducted and why you used the method (s).   You provide p values but don’t say what the chi square value is (I presume that is what you are mostly using??)

2)      In all your tables you should be detailed in describing what the table is showing.  If the estimates are weighted, this should be made clear and unweighted shown as well.  Also you need to provide confidence intervals. This is standard practice and will make this paper much stronger.  For example all your estimates of expenditure – are they means? Medians?

3)      Keep results as results – for example, lines 173 174 should be in the analysis section (presently missing) and not the results.

Other points to address:

4)      Avoid using value laden language such as in lines 271-273 which should be omitted (it isn’t necessary) or reworded in a neutral fashion- also lines 283-286 need to be reworded to a less exaggerated language particularly as it is an inference.

5)      You have evidently gone to a great deal of trouble to conduct this study but spend all your discussion saying why it wasn’t very good.  Please first of all, summarise what you did find and how that assists policy development for food insecurity.

6)      You could, and I thing should, align your work to other work done internationally with high income countries.  The subject matter is international and therefore use what you did find to align with other studies with similar finding to speculate what might be happening.

7)      Suggest possible further research based on this.

Author Response

Dear Reviewer 

Thank you for your time and consideration of our manuscript. I have appended our responses to all three of our reviewers so that you can see the genesis for all the changes we have made our manuscript.

Best wishes

Flora Douglas (on behalf of all authors)

Reviewer 2

 Can   secondary data tell us about household food insecurity in   a high-income country context?

Thank   you for allowing me to review this contribution to the Scottish food   insecurity knowledge base. I think that this paper is important and provides   a great deal of information, which may not be directly evidence of food   insecurity, but is certainly evidence of hardship and patterns of consumption   associated with that.

We are similarly grateful for   your time spent reviewing our paper and appreciate your assessment and   observations

I   am very puzzled by the conclusion as it doesn’t really fit with the analysis   or the title.  First of all, the use of the data in the study confirms   many other study findings and is a valuable source of information about low   income households in Scotland – so in that sense yes it can tell us something   about food insecurity if we accept the premise that there is a relationship   between the two (lots of literature on this).

While   the definition of what makes a food insecure household may not be broad   enough as it is not only low income that determines food insecurity, there is   plenty of good information in this paper that suggests they are tapping into   food insecurity (which the authors briefly reference in their introduction   and discussion).

We have altered the title to try   to address this concern and hope that this modification and the further   additions and changes made to our paper in the round, that are detailed   below, will help address this concern. We hope these changes will assure the   reviewer that our analysis provided a sense (“a partial picture”) of the nature of food insecurity in Scotland   (as the reviewer him or herself is saying our analysis taps into), but that   it revealed the challenge that exists in characterising and monitoring that   more multi-faceted experience of food insecurity, including its psychosocial   aspects which we argue are as important to monitor and address. We do accept   that there are other factors that can affect food and nutrition security   related to things like the availability, stability and usability of a food   supply, household food insecurity, but would also contend that in a high   income country context HFI is more generally associated with having   insufficient household income to achieve the food security, that is   predicated on there being sufficient, safe and stable food supply at the   national level. (Please see our additions to the introduction set out in 5.   Below supporting our argument).

We also accept that there are a range of   different definitions of household food insecurity, the Radimer conceptual   definition that underpinned our study (that was a given from the national   health promotion agency that commissioned this study, and the one that is   currently used within public health policy in Scotland), also underpins FAO   Food Insecurity Experience Scale measure.   

The   introduction needs to be increased to summarise what is know from the   international literature or food insecurity in high income countries. This   paper is too narrowly focused on Scotland which is fine for the actual study   itself but the introduction should be a broad base of information about the   subject matter and it is not.  Increasing this will make the discussion   easier as it too lacks enough connection with what is already known and   focuses too much on just Scotland – the aim of research is to be able to be   generalised where possible and much of the international literature is.

 We agree with the reviewer’s   contention of the need to situate our findings in an international context. We   admit to being a little puzzled that the reviewer is of the view that we have   not attempted do this in the introduction of our paper. For example the   opening paragraph (lines 35-46 are exclusively focused on the international   literature associated with general characteristics of and negative health   outcomes associated with food insecurity in high income countries. However,   we have added the following text to this paragraph to give some specific   examples of HFI prevalence in high income countries, and this additional text   is highlighted in yellow below.

Household food insecurity   (HFI) is a common problem for low income households in high income countries [1-4].  HFI exists when a household experiences the   inability “to acquire or consume an adequate quality or sufficient quantity   of food in socially acceptable ways, or the uncertainty that one will be able   to do so”[5]. It has been empirically   established to manifest itself involuntarily across four dimensions: (i)   quantity and (ii) quality of food; (iii) psychological impacts; and (iv)   socially unacceptable food and ways of obtaining food [6, 7][8]. In high income countries,   food insecurity is more commonly characterised by chronic compromises in   dietary quality and anxiety associated with accessing food, compared to acute   and chronic episodes of food deprivation, hunger, and starvation more   generally associated with low to middle income countries [8]. Critically, for health   and social care policy makers in high income country contexts, the experience   of food insecurity featuring poor diet quality leads to negative health   outcomes i.e. cancer, stroke, cardiovascular disease, diabetes and obesity,   and depression [8-13][14, 15]. Much   of the epidemiological evidence highlighting these associations have been   generated in the North American context where routine capture of HFI data has   taken place for some decades [16].   Nationally representative food security monitoring was established in the US   in 1995 with the highest recorded HFI prevalence observed during the global   recession of 2008-2009 (14.5% of the population) [17].   While there has been an observed decline in those figures, they remain in the   region of 12% of the population (15 million households) [18].   Canadian HFI prevalence runs roughly in line with US at 11-12% of the   population found to have been food insecure is food security monitoring was   established there 1994 [19].   HFI prevalence is also higher in specific population subgroups including   households with children, single‐parent households and indigenous, black and   Hispanic households [18, 20],   a pattern also observed in Australia and New Zealand [21, 22].   It is also widely argued that HFI is an outcome of income insufficiency in   relation to necessary household expenditures [20, 22-26]   and policy makers [27].

You   also have used innovative methods and this is a real contribution to the   knowledge base and may be able to be used (or adapted for use) by other   studies.  This is a positive part of your study and needs to be   acknowledged.

We have added the following to   the text in Section 4.1 paragraph 2 lines 301-304

Overall,   the indicators of HFI and the subsequent analysis used in this study are   innovative and could be adapted in other studies to bridge the gap in the   literature.

The   discussion appears to be more about what is lacking and use as a platform for   reform than what was added to the knowledge base.  The way in which this   is written begs the question why do it at all if it is so inadequate??    Obviously this in not what the authors meant (I hope).  I would suggest   adding at least a paragraph showing just how much extra information they have   successfully put together about 1) low income households in Scotland and 2)   their consumption habits.

We argue that the story of our findings is presented   on page 8 Section 4.1 lines 280-304, but have added some text (see tracked   changes) that hopefully makes this clearer.  

The   biggest omission and one that absolutely must be addressed is around the lack   of any explanation of the statistics used in the production of the   results.  The methods used were innovative and done well but where is   the information about the statistical analysis?  The following   additional information is required:

We hope the following   modifications (in comments 9-10 below) will satisfy the Reviewer that this   query has been addressed.  

1)      While   the description of how the measures were developed is excellent it is not   statistical analysis.  Please say what statistical analysis you   conducted and why you used the method (s).   You provide p values   but don’t say what the chi square value is (I presume that is what you are   mostly using??)

1.       Section 2.2. was amended as   follows:

Using   both LCFS and SHeS, the second stage involved estimating the numbers of   households at risk of being in food poverty, and investigating how prevalence   had changed over time. Using LCFS, this stage also included analysis of food   expenditure (£) and food-to-income shares (%) between lower and higher income   households. We also calculated the mean weekly expenditure on   food and non-alcoholic drinks, and their corresponding values in terms of   percentage by food group based on the Eatwell Plate. Two-tailed   independent t-tests were used to compare mean food expenditure (overall and   by food group) and food-to-income shares between HBAI and non-HBAI, and to   compare percentage DQI score between HBAI and non-HBAI and over time.

2)      In   all your tables you should be detailed in describing what the table is   showing.  If the estimates are weighted, this should be made clear and   unweighted shown as well.  Also you need to provide confidence   intervals. This is standard practice and will make this paper much   stronger.  For example all your estimates of expenditure – are they   means? Medians?

2.       The manuscript was amended as   follows:

Weight and sources of dataset are below each table/figure.

Mean has been added to the title of Tables 1 and 2 to make it   clearer what we calculated.

Confidence   interval and p-values were added were in Tables 1 and 2, where appropriate.

3)      Keep   results as results – for example, lines 173 174 should be in the analysis   section (presently missing) and not the results.

3.       We have followed the journal’s   guidelines regarding the naming of the naming of sections and have used the   so-called Materials and Methods section to present our account of the   analytical methods used. We have moved the section highlighted by the   Reviewer and moved it to page 3 lines 134-138.

Other   points to address:

4)      Avoid   using value laden language such as in lines 271-273 which should be omitted   (it isn’t necessary) or reworded in a neutral fashion- also lines 283-286   need to be reworded to a less exaggerated language particularly as it is an   inference.

4.    We are grateful to the reviewer   for pointing out this weakness in our discussion and hope that     the changes we have made to the language   used in this section will address this concern.

5)      You   have evidently gone to a great deal of trouble to conduct this study but   spend all your discussion saying why it wasn’t very good.  Please first   of all, summarise what you did find and how that assists policy development   for food insecurity.

5.    We would like to highlight the   changes we have made to page 282-332 where we placed the summary of our   results and hope our modifications makes our summary clearer, and in page 9   lines 366-376 and lines 383-384  where   we make it clearer what we think are the major strengths of this work. In   terms of the policy implications arising please see additional text on (page   10/11 lines 413 – 439 and page 11 lines 455-470.)

6)      You   could, and I thing should, align your work to other work done internationally   with high income countries.  The subject matter is international and therefore   use what you did find to align with other studies with similar finding to   speculate what might be happening.

6.    Please see comments to Reviewer 1   in 5. above. We hope our response here addresses the reviewer’s request.

7)      Suggest   possible further research based on this.

7.    Please see our comments   and added text in 16 and 17 below.

Reviewer 3 Report

Comments to the Author

Review report for International Journal of Environmental Research and Public Health

Title: Can secondary data tell us about household food insecurity in a high-income country context?

I think this is an interesting piece of work and is ultimately publishable but it need some minor revisions.  At present, the paper, in essence, argues that the measures based on the secondary data only capture the partial picture of household food insecurity (HFI).  A natural question is that, for governments, how to accurately identify those vulnerable people who are really food insecure, if we cannot fully draw on these secondary data.

As we know, HFI encompasses different dimensions (including prevalence, chronicity and severity) and also has different definitions. For example, the USDA classifies household into different HFI groups, ranging from food secure, low food secure to very-low food secure (based on 18 food insecurity questions). However, the definition in Europe is mostly defined as the prevalence of the household’s inability to afford meat/fish/poultry (or a vegetarian equivalent) every second day. I just think, is that possible to follow USDA’s criterion in future? In addition, as FAO has emphasized, HFI may be chronic, seasonal or transitory. For policy implications, what can we really learn from this study? It would be great this study could mention some future research directions, in particular, the measurement of HFI.

For limitations, it is better for authors mention them in the main text. First step is to establish a comprehensive measure of HFI. Then we might think what drivers account for the changes of HFI over time (in this study, a decline is observed)? Relating to this, getting better longitudinal data at the micro-level is quite important when evaluating the long-term effects of demographic and socioeconomic characteristics on the temporal variations of HFI (though it is beyond the scope of this study). In addition, this study only adopts a unidimensional indicator for HFI (though using different measures), it is necessary to use comprehensive measures of HFI such as USDA criterion in the US to assess the prevalence, chronicity and severity of HFI. Since geographical HFI differences might be driven by state-level or region-level institutional and social protection factors, unfortunately, it seeming missing in this study. The population ageing could be also a challenge for Scotland. How about the HFI situation for the elderly in this country? Lower socioeconomic status (SES) does not only encompass lower level of income. How about education or occupational status?  

Finally, some typos are identified in the main text. For instance, (£53.85 (29%) and £86.73 (15%) in the Abstract. Another is “[6,17,18[22]]” (p.2). Before formal resubmission, please avoid these typos.

Author Response

Dear Reviewer 

Thank you for your time and consideration of our manuscript. I have appended our responses to all three of our reviewers so that you can see the genesis for all the changes we have made our manuscript.

Best wishes

Flora Douglas (on behalf of all authors)

Reviewer 3

I   think this is an interesting piece of work and is ultimately publishable but   it need some minor revisions.  At present, the paper, in essence, argues   that the measures based on the secondary data only capture the partial   picture of household food insecurity (HFI).  A natural question is that,   for governments, how to accurately identify those vulnerable people who are   really food insecure, if we cannot fully draw on these secondary data.

1.       We have added the following text   to page 11 lines 455-470 to address this comment and that of Reviewer 2 in   15. above. We hope this modification addresses each reviewers’ point.

We contend that routine HFI monitoring   and evaluation endeavors must therefore also include direct qualitative   engagement with those highly vulnerable groups and the services and/ or   agencies responsible for their care and support, to inform and evaluate   social, economic and health policy changes intended to address HFI [73]. In addition, whilst periodic   episodes of absolute food deprivation are a public health and ethical concern   of any country, it is also the case that the less severe experience of food   insecurity resulting in chronic dietary quality compromises over time, is   likely to be the more prevalent experience in high income countries like   Scotland. This is an important distinction to be able to capture as   accurately as possible as chronic food insecurity experience is considered to   be as damaging to health and well-being over the long term as periodic   deprivation experiences [68] and this phenomenon needs more   research attention in HFI monitoring work in the UK and elsewhere than it   currently receives. Related to this is the need in the UK to develop a better   understanding to the impact of food insecurity on chronic or long-term health   condition management in the UK [73]. Having a clearer picture of the   severity and chronicity of people’s HFI experience as well as national and subnational   prevalence would assist researchers and policy makers to develop more insight   into this overlooked health care issue, and, create better informed policy   responses to address HFI in the Scotland and elsewhere in the UK.

As   we know, HFI encompasses different dimensions (including prevalence,   chronicity and severity) and also has different definitions. For example, the   USDA classifies household into different HFI groups, ranging from food   secure, low food secure to very-low food secure (based on 18 food insecurity   questions). However, the definition in Europe is mostly defined as the   prevalence of the household’s inability to afford meat/fish/poultry (or a   vegetarian equivalent) every second day. I just think, is that possible to   follow USDA’s criterion in future? In addition, as FAO has emphasized, HFI   may be chronic, seasonal or transitory. For policy implications, what can we   really learn from this study? It would be great this study could mention some   future research directions, in particular, the measurement of HFI.

2.       We agree with the Reviewer’s   argument that the HFI has different dimension that includes prevalence,   chronicity and severity and that these have important implications of public   health and health care policy in terms of the observed associations that have   been established in the US and Canada.. We also agree that this measure   provides more useful data for population health monitoring purposes compared   to the European one.

In answer to the Reviewer’s question about   whether it is possible to follow USDA’s criterion, as we have highlighted in   our first submission, policy and political decisions have been taken in   Scotland to pursue food insecurity monitoring using a derivative of the USDA   FIES. So it is currently policy here to do this.   Taking on board this point and the   reviewer’s further questions and observations in this comment, we have modified   the existing text to address their points. See below.  We hope these additions (starting at page   10 line 413) will satisfy the reviewers that we are taken on board their very   useful and highly relevant points.

Therefore, it has been encouraging to witness the   discussion and policy shifts in recent years highlighting the need for   routine monitoring of HFI across the UK. However, there is still no agreement   about the means and measures by which this should be done, with policy   differences emerging within the different nations of the UK regarding these [91, 92]. The Scottish Government   have recently accepted the main recommendations of the Independent Working   Group on Food Poverty in Scotland [93] and have introduced a   HFI measure which is a derivative of the UN / USDA Food Insecurity Experience   Scale [94] into the SHeS. The SHeS   operates on a continuous annual reporting basis, and provides sufficient data   for each individual health board area in Scotland in order to better   understand their respective population health dynamics over time. The   benefits of placing this measure here, and with it the potential routine   capture the more multi-dimensional HFI experience, are manifold. Firstly, it provides   the facility to assess and monitor food insecurity experience for different   subgroups (e.g. geographic location, age, ethnicity, household type,   occupational status and health status). Secondly, embedded HFI monitoring in   such a survey enables data linkage with other population data sets including   disease registers, and therefore enables better scrutiny of the impact of   food insecurity experience on population health outcomes [95]. Thirdly, the inclusion   of a such HFI measure also provides the facility to monitor prevalence and   severity (if not chronicity over time), and with it, the potential to develop   better understanding about the role different HFI experiences (in terms of nature   and severity) has in relation to health outcomes within the Scottish   population, something that was beyond the scope of our study. Fourthly, it   should also provide a robust means to determine the effectiveness of policy   interventions intended to address HFI. Indeed, it is important to stress the   benefits of introducing and retaining such a measure compared to the HFI   indicator used in Europe, which uses a more unidimensional indicator that is   based primarily on the prevalence of the household’s inability to afford   meat/fish/poultry (or a vegetarian equivalent) every second day [25]. Fifthly as the SHeS   survey routinely also captures household income data, it should be possible   to monitor and model changes in HFI prevalence in the context of changing   national and household economic circumstances and social policy changes,   something one-off cross-sectional studies cannot undertake.

3.       In relation to request to   highlight more information about research directions please see response in 16.   and 17. above.

For   limitations, it is better for authors mention them in the main text. First   step is to establish a comprehensive measure of HFI. Then we might think what   drivers account for the changes of HFI over time (in this study, a decline is   observed)? Relating to this, getting better longitudinal data at the   micro-level is quite important when evaluating the long-term effects of   demographic and socioeconomic characteristics on the temporal variations of   HFI (though it is beyond the scope of this study). In addition, this study   only adopts a unidimensional indicator for HFI (though using different   measures), it is necessary to use comprehensive measures of HFI such as USDA   criterion in the US to assess the prevalence, chronicity and severity of HFI.   Since geographical HFI differences might be driven by state-level or   region-level institutional and social protection factors, unfortunately, it   seeming missing in this study. The population ageing could be also a   challenge for Scotland. How about the HFI situation for the elderly in this   country? Lower socioeconomic status (SES) does not only encompass lower level   of income. How about education or occupational status?  

4.       We are grateful for the suggested   additions to the limitations section of our paper. They have been very   helpful in reminding us about some issues that were either missing from our   assessment of those or were not sufficiently evident and discussed in the   first iteration of the paper. We hope the changes to the text set out in 17.   provides further detail and is structured in a manner that satisfies the   reviewer that we have addressed his or her concerns here.

Finally, some typos are   identified in the main text. For instance, (£53.85 (29%) and £86.73 (15%) in   the Abstract. Another is “[6,17,18[22]]” (p.2). Before formal resubmission,   please avoid these typos.

5.       We have thoroughly checked the   paper again and have spotted and sorted a number of minor typographical   errors that the reviewer quite correctly has drawn our attention to.